# A Current Encyclopedia of Bioinformatics Tools, Data Formats and Resources for Mass Spectrometry Lipidomics

**DOI:** 10.3390/metabo12070584

**Published:** 2022-06-23

**Authors:** Nils Hoffmann, Gerhard Mayer, Canan Has, Dominik Kopczynski, Fadi Al Machot, Dominik Schwudke, Robert Ahrends, Katrin Marcus, Martin Eisenacher, Michael Turewicz

**Affiliations:** 1Forschungszentrum Jülich GmbH, Institute for Bio- and Geosciences (IBG-5), 52425 Jülich, Germany; 2Institute of Medical Systems Biology, Ulm University, 89081 Ulm, Germany; gerhard.mayer@uni-ulm.de; 3Biological Mass Spectrometry, Max Planck Institute of Molecular Cell Biology and Genetics, 01307 Dresden, Germany; canan.has@centogene.com; 4University Hospital Carl Gustav Carus, 01307 Dresden, Germany; 5CENTOGENE GmbH, 18055 Rostock, Germany; 6Department of Analytical Chemistry, University of Vienna, 1090 Vienna, Austria; dominik.kopczynski@univie.ac.at (D.K.); robert.ahrends@univie.ac.at (R.A.); 7Faculty of Science and Technology, Norwegian University for Life Science (NMBU), 1433 Ås, Norway; fadi.al.machot@nmbu.no; 8Bioanalytical Chemistry, Forschungszentrum Borstel, Leibniz Lung Center, 23845 Borstel, Germany; dschwudke@fz-borstel.de; 9Airway Research Center North, German Center for Lung Research (DZL), 23845 Borstel, Germany; 10German Center for Infection Research (DZIF), TTU Tuberculosis, 23845 Borstel, Germany; 11Center for Protein Diagnostics (ProDi), Medical Proteome Analysis, Ruhr University Bochum, 44801 Bochum, Germany; katrin.marcus@rub.de (K.M.); martin.eisenacher@rub.de (M.E.); 12Faculty of Medicine, Medizinisches Proteom-Center, Ruhr University Bochum, 44801 Bochum, Germany; 13Institute for Clinical Biochemistry and Pathobiochemistry, German Diabetes Center (DDZ), Leibniz Center for Diabetes Research at Heinrich-Heine-University Düsseldorf, 40225 Düsseldorf, Germany; 14German Center for Diabetes Research (DZD), Partner Düsseldorf, 85764 Neuherberg, Germany

**Keywords:** lipidomics, bioinformatics, data format, database, mass spectrometry, standardization, FAIR

## Abstract

Mass spectrometry is a widely used technology to identify and quantify biomolecules such as lipids, metabolites and proteins necessary for biomedical research. In this study, we catalogued freely available software tools, libraries, databases, repositories and resources that support lipidomics data analysis and determined the scope of currently used analytical technologies. Because of the tremendous importance of data interoperability, we assessed the support of standardized data formats in mass spectrometric (MS)-based lipidomics workflows. We included tools in our comparison that support targeted as well as untargeted analysis using direct infusion/shotgun (DI-MS), liquid chromatography−mass spectrometry, ion mobility or MS imaging approaches on MS^1^ and potentially higher MS levels. As a result, we determined that the Human Proteome Organization-Proteomics Standards Initiative standard data formats, mzML and mzTab-M, are already supported by a substantial number of recent software tools. We further discuss how mzTab-M can serve as a bridge between data acquisition and lipid bioinformatics tools for interpretation, capturing their output and transmitting rich annotated data for downstream processing. However, we identified several challenges of currently available tools and standards. Potential areas for improvement were: adaptation of common nomenclature and standardized reporting to enable high throughput lipidomics and improve its data handling. Finally, we suggest specific areas where tools and repositories need to improve to become FAIRer.

## 1. Introduction

Mass spectrometry (MS) is a state-of-the-art analytical technology, which enables the rapid and consistent identification and quantification of lipids in lipidomics, metabolites in metabolomics and proteins in proteomics for biomedical and biochemical research purposes [1]. Through the technological advances achieved during the past twenty years, main performance parameters were improved, such as mass accuracy and sensitivity. MS has become the analytical method of choice for many omics disciplines. All MS-based omics technologies share the following general workflow: (i) sample separation, (ii) analysis by a separation technology such as liquid chromatography (LC), hydrophilic interaction liquid chromatography (HILIC), reversed phase liquid chromatography (RPLC), supercritical fluid chromatography (SFC), gas chromatography (GC) or capillary electrophoresis (CE), (iii) mass spectrometric measurements supported by different ionization principles, e.g., via electrospray (ESI), electron ionization (EI), desorption electrospray ionization (DESI) for ‘matrix-assisted laser desorption and ionization’ (MALDI), (iv) separation and detection of the ions by the *m/z* values in the mass analyzer applying several physical principles and (v) storage of MS spectra, where the signal intensities are proportional to the abundance of the molecular species. However, applied omics workflows are comprised of several specific customizations to be well suited for the investigated biomolecule class and the associated analytical question.

Lately, ion mobility spectrometry (IMS) has gained a lot of attraction as a method of separating ions in the gas phase [2]. In IMS, ions are brought into interaction with an inert collision gas using static or modulated electric field gradient configurations to achieve ion separation and selection. An ion’s retention behavior in the IMS separator is determined by its average rotational collisional cross section (CCS), such that more compact ions tend to migrate faster toward the outlet of the IMS separator by exhibiting fewer collisions. Further, its behavior is influenced by the interaction of the ion with the superimposed electric field and effective waveform, which can either filter (FAIMS) ions with specific mobility, separate ions in an electric field gradient within a drift tube (DTIMS) or separate ions into ion packets by a traveling wave electric field within stacked ring ion guide (TWIMS).

For the further characterization of a given molecule in a targeted lipidomics workflow for the validation and quantification of lipids, specific precursor *m/z* values and select potential fragment *m/z* values (transitions in an inclusion list) are tracked using robust and comparably inexpensive triple-quad MS instruments in selective reaction monitoring (SRM) mode, which allows for the identification and quantitation of lipids on the class level. Orbitrap-type or time-of-flight (TOF) MS instruments with a higher mass resolution and the ability to perform a full-scan acquisition in parallel reaction monitoring (PRM) mode for selected precursors, measuring all fragment ions simultaneously, can be used for targeted lipidomics to achieve a deeper MS fragment coverage, allowing for species or subspecies identification.

In untargeted lipidomics workflows for discovery applications, no previous inclusion list is provided, thus requiring MS instruments that can operate in a data-dependent acquisition (DDA) or data-independent acquisition (DIA) mode to obtain a full-scan precursor and fragment mass spectra of either top-k *m/z* signals with the highest intensity or all ions contained in predefined *m/z* windows. Such experiments are often performed on instruments with high mass resolutions to further reduce ambiguities caused by isobaric lipids.

Tandem mass spectrometric experiments (MS^2^) are applied to gain further insights into the lipid structure and various fragmentation methods are applicable to record precursor-specific fragment spectra. However, collision-induced dissociation (CID) is the most widely established approach. Identification software is applied to identify molecules by comparison of generated MS^2^ spectra with theoretical fragment spectra or with reference spectra from a database. The quantification of molecules is usually performed using the corresponding precursor mass spectra but may also be performed on selected MS^2^ fragments. Higher-level fragmentation series for identification and quantification are also applicable, where the mass spectrometer selects MS^2^ fragment ions for further fragmentation (MS^n^). Finally, the resulting data, i.e., raw MS^1^ and MS^n^ spectra and chromatographic retention time (RT), drift time or collisional cross section, scan polarity, collision energies and corresponding metadata such as MS device settings, are stored in vendor-specific data formats.

Current data formats, associated metadata and software tools face the challenge of keeping pace with technological developments. In addition to the respective specifications of the various mass spectrometry workflows described above, aspects of standardization, data management and software compatibility must also be considered. One of the largest challenges in current science [3] is to keep informatic pipelines sustainable and reproducible. Software with an available source code, optimally under a permissive open-source license, ensures that analyses performed today can, in theory, be reproduced in the future. Furthermore, software maintenance and development are easier to achieve with open-source software. Contributors from the community support further validation and development with greater ease and a lower entry barrier.

For readers who prefer a more in-depth review of analytical lipidomics methods, associated (bioinformatics) challenges and best practices, we would recommend [1,4,5,6]. For a comprehensive review of metabolomics software and resources, we recommend [7] and [8] for software and libraries written in the programming language R.

## 2. Materials and Methods

With this review, we want to provide a comprehensive and up-to-date review of the software, tools, databases and other resources connected to processing lipidomics data from mass spectrometry experiments. We thus searched for the terms, “lipidomics software”, “lipidomics tool” and “lipidomics database” in PubMed and Google Scholar and selected references that were either published between 2016 and December 2021 or were available as preprints as of December 2021 that were associated with mass spectrometry for lipid identification and/or quantification. We opted to include software and other resources from the past fifteen years if they are still being maintained and updated, focusing on tools that are either freely available to academic users and/or that publish their source code under an open-source license. We also included software and resources for metabolomics data when their utility for application to lipidomics data was apparent. We summarize the selected resources, to the best of our knowledge, within Appendix A. We also provide this table via the GitHub repository at https://github.com/lifs-tools/awesome-lipidomics (accessed on 24 May 2022).

Finally, we review and discuss the current status of standardization in data formats and reporting conventions in lipidomics to point out potential areas of improvement. This includes the question to which extent the standard formats, initially developed by the Proteomics Standards Initiative within the Human Proteome Organization (HUPO-PSI) [9] for proteomics data and later made usable for metabolomics and lipidomics data, are already applied for lipidomics data. We hope that these may be picked up by the Lipidomics Standards Initiative (LSI) [10] or other interested parties to further improve interoperability between lipidomics tools and resources and with other tools and resources from other omics disciplines.

## 3. Data Standards and Formats

Most vendors of mass spectrometers use proprietary or non-standard data formats for MS data, complicating reusability, interoperability, results comparison and data exchange (see Figure 1). Additionally, many software-specific file formats aggravate this problem, especially if only in-house developed file converters without regular updates are available. Thus, the use of standardized data formats, vocabularies and ontologies is indispensable to ensure the reusability and interoperability of scientific data for both humans and machines, as formulated by the FAIR Guiding Principles for scientific data (i.e., data should be findable, accessible, interoperable and reusable) [11]. Data adhering to the FAIR principles can be found in public repositories such as PRIDE for proteomics [12] and MetaboLights [13] or Metabolomics Workbench [14] and is discoverable in cross-omics resources such as the Omics Discovery Index (OmicsDI) [15]. The FAIR principles facilitate the interoperability of software tools within data analysis pipelines. Consequently, the efficiency of bioinformatics infrastructures and biomedical research can dramatically improve by following these guidelines [16]. Thus, in summary, data standardization and providing fully up-to-date and maintained converters are a crucial task in the computational mass spectrometry field [17].

To address these challenges for proteomics, the HUPO-PSI has been active since 2002 in the definition of minimum information requirements [18], standard formats [19] and ontologies [20]. The HUPO-PSI defined XML-based standard formats, such as mzML [21,22], for the vendor-neutral representation of raw mass spectrometer output (raw spectra, chromatograms, peak lists), mzIdentML [23,24] for peptide and protein identification results and mzQuantML [25] for quantification results. For MS imaging data, the imzML format was developed [26] by the Mass Spectrometry Imaging Society, but in close alignment with the metadata structure of the mzML format. All HUPO-PSI formats can be annotated semantically by controlled vocabulary (CV) terms that are defined in ontologies such as the mass spectrometry CV [27]. By defining mapping rules that describe which CV terms are allowed at which position in a data file, a semantic validation of that data file by dedicated validation programs is possible [28]. The XML-based formats are basically both human- and machine-readable but lack usability with generic text processing or spreadsheet software. Thus, scientists requested a more human-readable, editable and platform-independent file format for the resultant files of a proteomics investigation. This was realized in the tabulator-separated format, mzTab [29]. It allows for the storing of both identification and quantification results in an Excel-compatible spreadsheet format while still adhering to a pre-defined but extensible overall structure that is enriched using CV terms and semantic constraints that allow for computerized parsing and validation.

In the last decade, the metabolomics standards initiative (MSI) [30] has already defined minimum information guidelines [31] and initiated a standardization process that is based on the PSI standards [32]. As a result, the MSI has established important extensions to the PSI data formats, such as the support of GC-MS data in mzML carried out by the COSMOS project [33] and including missing CV terms into the PSI-MS ontology. Moreover, mzTab was adapted to fully support metabolomics and lipidomics data from mass spectrometry experiments in the mzTab-M 2.0 format [34]. Analogously, in 2018 a group of lipidomics experts with experimental and bioinformatics backgrounds founded the Lipidomics Standards Initiative (LSI), cooperating closely with other societies as well as with the PSI in order to refine updates to the most relevant PSI standards (e.g., ontologies, controlled vocabularies, data formats) for better reporting of lipidomics data.

These new data formats for results reporting need to follow a well-defined structure, defined by a computer-readable and validatable schema. Typically, a distinction is made between required, recommended and optional information that data curators of such a file must, should or may include. All of the following formats have in common that they are based on a tabular, human readable and easily inspectable data model, consisting of linked tables that report study and sample metadata, quantities, features, identification details and supporting evidence, either in a single file (mwTab [35], mzTab and mzTab-M) or as separate files (ISA-Tab [36]). The generation of report files following these formats is supported by specifications, supplied validation tools and libraries in different programming languages to simplify implementation [35,37,38,39]. First, recommendations for minimum reporting standards for lipidomics mass spectrometry have been published [40], which align well with the supported metadata in the above-mentioned data formats.

## 4. Software for Lipid Identification from Mass Spectrometry

The recent development of lipid identification tools has aimed to propel the rapidly emerging field of lipidomics by improving the quality and performance of applied algorithms, while integrating novel separation techniques and high-resolution mass spectrometers. We reviewed a total of 31 openly available software tools for lipidomics data processing and identification that were published between 2006 and the end of 2021. We evaluate the usability of common data formats and, specifically, of PSI standard data formats as either input or output formats and their support for at least one of the lipidomics workflows (see Table 1 for reference). A full list of these tools and their supported input, output and configuration formats is provided in Appendix A.

We categorized the tools by supported workflow (targeted, untargeted or both), sample handling (separation, e.g., chromatography, ion mobility, direct infusion, imaging), MS level, summarizing targeted, selected ions under MS^1^, MS^2^ for shotgun and DDA approaches and MSE/DIA for data-independent approaches, based on their own claims in their primary publications or documentation.

Concerning lipid identification, we broadly distinguish between tools that use either a rule-based or a library-based identification approach.

Rule-based tools must describe at least precursor ion *m/z*, MS^2^ fragments and (relative) fragment intensity ranges for lipid class, species or subspecies identification. In order to reduce the chance for false-positive identifications, these approaches often also apply further validation rules, such as fragment signal intensity ratios that must fall within certain bounds. However, these rule-based approaches can be customized to also allow for identification on a more precise lipid structure level if the necessary data is available. In principle, these approaches are very flexible and allow for the query of spectra for certain patterns that are indicative of specific lipid species. This makes them applicable to targeted, as well as untargeted, analysis.

Library-based approaches use either in-silico generated MS^2^ spectra for lipids derived from their structural representation or experimentally acquired and post-processed spectra. To assign a putative identity to measured lipid mass spectra, a variant of the dot product score or other related vector scores is often used [41,42].

We further indicate whether tools support quantitative output, such as intensities, areas, relative or absolute quantities or if they only support qualitative lipid identification output. For these tools to be included in larger processing workflows, the supported data formats for input and output are crucial. In the mass spectrometry and lipidomics field specifically, we can distinguish between text (human readable) and binary file formats. The latter are often the raw data vendor formats, but can also include local database files, such as the blib format for mass spectral libraries or the common sqlite database format. Within text-based formats, we can distinguish structured ones that follow a specific schema for MS data, such as the Mascot Generic Format (MGF), NIST Mass Spectrum format (MSP), MS2 [43] or mzTab(-M) and semi-structured ones, such as CSV, JSON or XLSX, where the latter is a compressed XML format. XML-based formats are well-adapted to be machine readable and validatable and are used in the PSI format mzML, as well as its predecessors, mzXML [44] and mzData [45]. TXT formats are generally only weakly structured but remain human-readable.

Maintenance, accessibility and reusability are important factors in being able to create and maintain reproducible processing pipelines from openly available tools. We therefore also captured the date of the last release for each tool with a granularity of one year and whether it is available under an explicit open-source license, and if so, under which one specifically. This is also an important aspect for the original authors of a tool, as sustainable development and maintenance of bioinformatics software through a lack of continued funding is still an issue. Open access to the software can help in building up a community around it, where maintenance and further development can be shared between different stakeholders. We did not specifically record whether a tool’s source code is available via a source code repository platform such as GitHub or GitLab, but generally recommend that for open-source software, since these platforms will make the source code available for the foreseeable future.

Lastly, we list the programming languages that were used to develop the tool. This can have an impact on operating system platform independence and may make reuse of the software easier for certain user demographics, e.g., MS EXCEL and VBA macros may simplify usage by non-bioinformaticians but have clear limits to the Windows platform and limit integrability into non-UI driven workflows.

### 4.1. Targeted Workflow

LIMSA [46,47] supports data from both LC separation, as well as direct infusion workflows. In a first step, vendor data needs to be converted to the NetCDF format using the authors proprietary but free of charge tool, SECD, which is then used to export MS data to LIMSA via EXCEL. LIMSA itself is implemented in C++ as an EXCEL add-in and provides peak finding, identification, isotopic correction and absolute quantification based on calibration lines and labeled internal standards. Unfortunately, we were not able to find a publicly available version of the software.

LipidomeDB [48,49] is a web application for the processing of direct infusion and differential ion mobility MS lipidomics data. It requires a user login but is otherwise free to use. LipidomeDB supports isotopic correction and absolute quantification via class-specific labeled lipid standards and linear calibration curves. Input data needs to be provided in XLSX format and can be exported after identification and quantification as XLSX and HTML.

LipidQuant [50] is a tool for quantitative lipidomics in lipid class separation workflows, such as HILIC or SFC coupled to MS, based on EXCEL and Visual Basic for Applications (VBA). It supports input of *m/z* and sample-wise quantity data in TXT or generally tabular formats from vendor software. It includes an extensible built-in database of lipid species, organized by lipid class, and performs type II isotopic correction and absolute quantification using class-specific, heavy labeled (deuterated) internal lipid standards. Output is available from the XLSX worksheet.

We describe tools that support both untargeted and targeted workflows in Section 4.3.

### 4.2. Untargeted Workflow

ALEX 123 [51] is an online database that provides comprehensive fragmentation information on 430,000 lipid molecules from 47 lipid classes across five different lipid categories. Output of ALEX 123 is provided in HTML format. In combination with LDA2, it was used for lipid and lipid fragment identification in LC-MS/MS data. Alternatively, ALEX [52] can be used for lipid identification on a species level from high-resolution FTMS data. The source codes of ALEX and ALEX 123 are not publicly available.

Greazy [53] is well-integrated with the ProteoWizard tool suite and supports both chromatography-MS as well as DI data. It generates a search space of phospholipids and theoretical MS2 spectra based on user input. Experimental MS2 spectra are searched against the phospholipids in the search space with adjustable precursor mass tolerance. The match score is computed based on a combination of hypergeometric distribution and intensity score, considering the number of observed fragments for each lipid. The lipid spectrum matches are filtered based on density estimation and the hits above the score threshold are reported in mzTab 1.0 format.

Lipid Data Analyzer 2 (LDA2) [52,53] supports untargeted LC-MS/MS lipidomics workflows and is implemented in JAVA. It accepts the following input formats for MS data: raw, .d, wiff, chrom and mzXML. It requires additional quantitation files (XLSX) with lipid class/species to mass/adduct mass association and additional expected RTs for each experiment. In LDA2, custom platform and ionization energy-specific fragmentation rule sets for lipid class and scan species level fragment identification can be defined. Identification and quantification results are stored in XLSX, CSV, mzTab 1.0 and most recently, mzTab-M 2.0.

LipidBlast [54,55,56] is a suite of XLSX/Visual Basic for Applications (VBA) macros that can generate in-silico tandem MS libraries for lipid identification with other tools, such as NIST’s MS Search application. Input formats are MSP, MGF and XLSX, while output can be generated in MGF and XLSX formats. It is not actively developed any longer, but its libraries have been integrated into MS-DIAL.

LipidDex [57] is also implemented in JAVA. It uses in-silico fragmentation templates and lipid-optimized MS^2^ spectral matching to identify and track lipid species in LC-MS/MS experiments. It can calculate peak purity and determine co-isolation and co-elution of isobaric lipids and is able to remove ionization artifacts. It reads data in MGF or mzXML formats and saves identification results in CSV tables.

LipidFinder [58,59,60] is a Python tool and web application available from the LIPID MAPS website that supports untargeted identification of lipids in LC-MS data, using XCMS for initial feature finding and custom filter and post-processing steps specifically tailored to lipidomics. Input formats are those that are also supported by XCMS, but specifically CSV and JSON, to transfer feature data and configuration settings to the application. LipidFinder supports the generation of reports in PDF, XLSX and CSV formats.

LipidHunter [61] identifications are based on (glycero-)phospholipidomics MS^2^ spectra measured by RPLC-MS/MS or direct infusion methods, integrating with LIPID MAPS for bulk lipid search. It supports mzML as an input format from LC-MS/MS and data-dependent shotgun acquisitions. Input files need to be split into an MS^1^-only file, covering survey scans for faster processing, and a complete file that contains MS^1^ and MS^2^ scans. LipidHunter extracts fragment ions based on a user-definable configuration and links MS^2^ fragment information to parent ions that are identified against the LIPID MAPS database. It finally performs a lipid species assignment based on their product ions and additional rules. LipidHunter reports quantification and identification results in HTML, CSV and XLSX.

LipidIMMS Analyzer [62,63] is a web application for lipid identification in chromatography ion mobility workflows. It uses an internal database of MS^1^, CCS, RT and MS^2^ information and applies a weighted composite scoring to assign the final identification. It accepts data in MSP and MGF formats and supports output in CSV and HTML.

LipidMatch [64] supports LC-MS, imaging and direct infusion workflows based on an extensive in-silico MS^2^ fragmentation library including 56 different lipid types. It uses a rule-based approach for lipid identification against the precursor and fragment *m/z* values, including definable adducts, and it is implemented in R. DDA as well as DIA data are supported through peak picking with tools such as MZmine or XCMS. LipidMatch accepts input in CSV (feature tables) or MS^2^ (MS/MS data) format and provides annotated and identified results down to the subspecies fatty acyl level. It exports identification results in CSV format. LipidMatch Flow converts vendor file formats with msConvert on the fly.

LipidMiner [65] supports LC-MS/MS DDA data and uses the LIPID MAPS structure database as its library for lipid identification using a rule-based approach. It is implemented in Python and C# and provides input from Thermo raw files. Output is provided in XLSX and CSV formats.

LipidMS [51] is an R package that supports the processing of high-resolution, DIA-MS data. Due to the missing direct relation between the precursor and fragments in DIA, the package applies a score to assess the co-elution of both for grouping, based on fragment and ion intensity rules that allow annotation on species, molecular subspecies (fatty acyl) and structural species (FA position) level. Input may be provided in mzXML or CSV. Output is available as R objects, which can be easily converted and exported into CSV and other tabular formats.

Lipid-Pro [66] is another tool that supports DIA LC-MS/MS data. Implemented in C#, it uses a lipid compound and fragment library and applies matching rules to identify precursor fragment associations based on retention time-aligned, pre-processed data. Input can be provided in CSV format, while output is available as XLSX or TXT.

LipidXplorer [67,68] supports DI-MS lipidomics workflows regardless of the lipid category, implemented in Python. It transfers filtered and averaged representative spectra (from all scans based on the measurement settings of the data) into a master scan. The master scan is then searched against the fragmentation rules per class and per mode as provided by query scripts written in Molecular Fragmentation Query Language (MFQL), which is inspired by the SQL database query language. The tool currently supports Thermo raw and mzML files as well as text file-based import (CSV for MS^1^ and DTA for MS2, in v1.2.7) as input files and generates comma-separated output files. The output file can be programmed by MFQL and usually reports lipid species found with mass, chemical formula, identification error, lipid name, isobaric species, if any, along with precursor and fragment ion intensities per sample (CSV).

LiPydomics [69] is a Python tool for HILIC ion mobility MS lipidomics data analysis. It uses a custom experimental database with *m/z* and CCS values for 45 lipid classes and HILIC retention times for 23 lipid classes. CCS prediction and HILIC retention time prediction for lipids that are not contained in the experimental database are realized by applying machine learning to the experimental database reference values. Identification is performed using a rule-based approach on *m/z*, RT and CCS values. LiPydomics accepts CSV files as input and provides results in XLSX format.

LIQUID [70] supports identification of lipids from LC-MS/MS experiments with a customizable library and adaptable scoring model that includes quartiles of fragment intensities. The library covers over 30,000 lipid targets in nine distinct lipid categories, 29 lipid classes and 85 subclasses, sourced from LIPID MAPS and extended with additional lipids. It is implemented in C# and supports input in Thermo Fisher raw format and mzML. Processing results can be exported in CSV, mzTab or MSP formats.

LOBSTAHS [71] is implemented in R for the identification of lipids, oxidized lipids and oxylipin biomarkers in LC-MS data. It uses XCMS and the R/Bioconductor package CAMERA [72] for feature detection and aggregation and validates potential lipid features against an internal *m/z* library of lipid species adducts using a rule-based approach based on adduct order of intensity. Input is therefore supported in all formats that XCMS supports. Output can be exported in XLSX or CSV formats.

For oxidized phospholipids, LPPTiger [60] is an option for data-dependent LC-MS/MS data. It is implemented in Python and uses in-silico generated spectral libraries together with a composite score based on individual similarity, rank, fingerprint, isotope matching and specificity scores. It reads data in mzML, XLSX and TXT formats as input (MSP for the library format) and outputs as XLSX and HTML.

MassPix [73] is an R library for the analysis of imaging-MS lipidomics data. It uses an MS^1^
*m/z* library for rule-based identification. It reads imzML format as input and annotates deisotoped *m/z* values against its internal generated library. Identified results can be exported in CSV format.

MS-DIAL 4 [74,75], written in JAVA, supports chromatography, CE and ion mobility workflows. It applies a spectral library search approach, based on a MS fragment library of 177 lipid subclasses. MS-DIAL 4 performs peak picking, alignment annotation and quantification. Identification combines scoring and a rule-based approach that is guided by a decision tree and provides different levels of confidence. As input formats, multiple vendor formats and mzML are supported, while outputs can be written in CSV, XLSX and mzTab-M. MS-DIAL also supports retention time prediction and offers comprehensive visualizations.

MZmine 2 [76] is a modular software for untargeted, chromatography-based metabolomics, with support for lipid species identification using spectral libraries and rules for annotation. It is implemented in JAVA and offers to read input from a variety of vendor formats as well as from open formats as input and it is also able to export identification and intensity data in common spreadsheet and tabular formats and supports mzTab for reading and writing. The upcoming MZmine 3 will also support mzTab-M.

XCMS [77,78] is a generic R/Bioconductor library for mass spectrometry feature finding and grouping and has no dedicated support for lipid identification. It uses a spectral library-based approach for feature identification, but other packages may provide other functionality more tailored for lipids. XCMS supports LC-MS/MS data in mzML, mzXML and netCDF formats and outputs feature tables in CSV, XLSX or other formats supported by the R ecosystem.

### 4.3. Targeted and Untargeted Workflow

The final batch of tools support the analysis of targeted and semi-targeted or untargeted lipidomics data.

LipidCreator [79,80], together with Skyline [81], is primarily designed for targeted lipidomics analysis, but through Skyline’s support for DIA analysis, can also be applied for untargeted workflows. LipidCreator is used to create transition lists and spectral libraries for more than 60 lipid classes, either using predefined libraries for common species and tissues or by manual selection of lipid classes, head groups and fatty acyl parameters. Transitions and a spectral library derived from the in-silico transition list can be transferred to Skyline to be used with its peak/transition detection and integration and its spectral matching features. All major vendor formats are supported, as well as mzML for input. Results can be exported in XLSX and CSV formats, while spectral libraries are exported in the open BLIB format.

LipidPioneer [82] is an EXCEL template implemented in VBA supporting more than 60 lipid classes, including oxidized ones. It allows the generation of custom lipid inclusion lists based on sum formulas of adduct masses for use in targeted and untargeted workflows. These can then be used by other software for lipid identification, such as MZmine, MS-DIAL or Greazy, or for Quality Assurance (QA) and Quality Control (QC) applications. LipidPioneer supports export in any format supported by EXCEL, e.g., CSV or EXCEL.

LipidQA [83] supports both targeted and untargeted workflows for DI-MS. It is implemented in Visual C++ and uses a fragment ion and lipid chemical formula database to perform spectral matching for identification. Absolute quantitation with calibration curves is also supported. LipidQA can read data in Thermo and Waters vendor formats and provides its results in CSV format.

LipoStar [84], implemented in C#, supports data from chromatographic separation and ion mobility for DDA and DIA workflows. It uses a compound and fragment library and rule-based validation for the identification of lipids. LipoStar reads vendor MS data and supports the exporting of results in the CSV format.

LipoStarMSI [85] is LipoStar’s sibling software for direct infusion and imaging MS lipidomics. It uses a spectral library and rule-based approach for lipid identification. LipoStarMSI is also implemented in C# and can read vendor formats of Bruker and Waters as well as the open imzML format. Output is exported in CSV format.

SmartPeak [86] uses OpenMS [87] at its core and supports absolute quantitation in targeted and semi–targeted workflows. It is implemented mainly in C++ and implements MRM-specific peak integration and feature selection on top of established OpenMS methods. SmartPeak’s primary input format is mzML, while transitions, parameters and sample sequence information are provided in CSV format. Results can be exported in mzTab, XML and CSV formats.

Smfinder [88] has parts that are implemented in Python and some parts that are implemented in R. It supports targeted, untargeted and 13C labeling workflows. Lipid identification is performed based on plausible sum formulas first, with subsequent validation using a spectral library. The untargeted workflow uses XCMS for feature detection. Smfinder supports mzML and mzXML as input data formats. Results can be exported in XLSX and TXT formats.

Out of the 31 tools for lipid identification we reviewed, 6 of 31 (>19%) did not provide a release version that could help to ensure reproducibility when authors want to compare their software to those of others. Eight of 31 tools (>25%) had no explicit license defined. Just as many, but not necessarily the same ones, did not provide the source code in an openly accessible way.

## 5. Data Post-Processing, Statistical Analysis, Visualization and Pathway Integration

Tools for lipidomics data post-processing, e.g., for absolute quantification, nomenclature standardization, statistical analysis, visualization and pathway integration, are important steps to integrate lipidomics MS data into a biochemical or medical context (see Table 2).

A lipid ontology, such as the biochemically inspired one of LIPID MAPS, should be a natural complement to chemical structure and function-based ontologies such as Chemical Entities of Biological Interest (ChEBI) [89], focusing on the taxonomic organization and classification of lipids by their functionalization and other molecular characteristics and then linking that information to other resources, such as the Gene Ontology (GO) [90,91]. Some attempts in this area have already been made by (semi-)curated ontologies such as the OWL-based LiPrO and its extension [92,93] and by LipidGO [94], which are, unfortunately, no longer available. Recently, automated approaches have been reported, e.g., for more generic molecules by ClassyFire [95] and in a more manual approach, also supporting enrichment analysis with Lipid Mini-On [96] and LION/web [97], but the momentum in this area has not yet led to a consensus and accepted reference ontology.

LipiDisease ranks the associations of lipids and diseases by mining PubMed records [98] based on their Medical Subject Headings Thesaurus (MeSH) annotations. Machine learning approaches in the area of lipid classification have also been developed to classify lipids based on sum formulas with SMIRFE [99] and based on SMILES [100]/SMARTS [101] structural representations in Lipid Classifier [102] against the LIPID MAPS structural ontology. BioPAN [103] is a web application for the exploration of mammalian metabolic pathways based on LIPID MAPS classification, including enrichment analysis and comparison between conditions.

A further challenge is the canonical naming of lipids. Most tools use the naming rules that were established at the time of their development. Thus, the most recent proposed lipid nomenclatures are usually not covered, and lab-specific dialects hamper the general re-use of such data. The authors of Goslin [104,105] provide libraries in multiple programming languages for automatic parsing and normalization of lipid shorthand nomenclatures based on context-free grammars, as well as a web application that provides mappings to LIPID MAPS [106] and SwissLipids [107] via the normalized lipid name as a lookup key. LipidLynxX [108] provides a web application and Python library for a similar use-case, based on regular expressions, and also supports mapping of lipid names to external databases via an online, federated search. RefMet [109] takes a more global approach and defines a common reference nomenclature for metabolomics, including lipids, and further offers a parsing and conversion service as part of LIPID MAPS and Metabolomics Workbench. Pauling and co-workers [51] proposed a nomenclature for fragment ions in lipid mass spectra that could be used to help describe fragment-based evidence for final lipid identifications and provide an online resource with common fragments for many lipid classes (see “Alex123” in Table 1).

LICAR [110] provides isotopic correction of lipidomics data that were acquired in targeted MRM mode after class-specific separation as a user-friendly R/Shiny web application.

The last category of tools and web applications provide statistical analysis, comparative analysis and comprehensive visualizations specialized for lipidomics. In this regard, lipidr [111] as an R library and LipidSuite [112] as a supporting R/Shiny web application provide assistance for interactive analysis and visualization. Specifically for the statistical analysis of fatty acid compositions from complex lipids, the liputils [113] package is available in Python, using the RefMet nomenclature as input. A more general solution for metabolomics data, with lesser support for lipid structural level-specific analysis, but in general many more statistical and machine learning methods for the analysis of untargeted data is the MetaboAnalyst web application [114] and supporting R library. As part of MZmine 2, Kendrick-referenced mass-defect plots [115,116] are a very helpful visualization to support the identification of lipid classes. LUX Score [117,118] provides a lipidome homology model calculated on the basis of a chemical space model that utilizes template SMILES of lipids as input. This enables it to distinguish, cluster and visualize qualitative changes in lipidome compositions between different tissues within and across species.

Out of the 16 tools in this category, 4 of 16 (25%) did not provide a release version that could help to ensure reproducibility when authors want to compare their software to those of others. Four of 16 tools (25%) had no explicit license defined. The source code was not available in an easily accessible way in 5 of 16 cases (>31%).

## 6. Databases, Repositories and Other Resources

Lipid databases and repositories (see Table 3) need to consider the currently incomplete structural resolution of mass spectrometry data. Biological samples contain a large structural variety of lipid classes and species, where especially the latter may not be represented in a database in their entirety. Even with high-resolution tandem mass spectrometry, the fatty acyl composition of lipids, e.g., the variety in fatty acyl chain length, saturation (number of double bonds) and other functionalizations does not allow identification on the lipid molecular species level. This effectively leads to ambiguous lipid identification, which is why chromatographic separation and lately, ion mobility, have been added to the analytical toolbox to improve specificity.

The high structural variety motivated the design of a tailor-made nomenclature for lipids, their structures and fragment ions. The first proposal for a unique definition and ontological classification of lipids was the nomenclature defined by the LIPID MAPS consortium and database [106,120,121,122], supplemented by SMILES string notation to represent fully resolved lipid structures with defined, uniform rules for the order of the head group and fatty acyls for the SMILES string generation.

The original LIPID MAPS nomenclature covers three main levels (category, main class and subclass) to broadly distinguish lipids that are reported on a structural level with full stereochemistry information. It therefore lacked concepts for describing further intermediate levels that are accessible with current MS technology with progressively more structural information. These levels have been incorporated by the more detailed nomenclature introduced by [123], which was prototypically implemented in the LipidHome database [124], which contains computationally generated structures for Glycerolipids and Glycerophospholipids. This hierarchy was further expanded in SwissLipids [107], enriched with information on experimental evidence and cross-links to biochemical reactions involving those lipids via Rhea [125]. The shorthand nomenclature was updated recently [126], with the changes being incorporated into LIPID MAPS successively.

The usage and report of Lipid identification criteria vary widely within the software solutions. Some tools report the actual fragment ions together with indicative rules that have led to the identification. At the same time, they allow the definition of a confidence level based on complementary structural information derived from MS^2^ in positive and/or negative ionization mode and/or other means of identification, such as structural knowledge about measured moieties. Especially in lipidomics where MS-based methods can resolve structural details only to a certain degree, providing evidence for the level of identification (e.g., lipid species, subspecies, fatty acid positions, isomer level, potential for isobaric species) is crucial to avoid overreporting, misinterpretation of results and to enable proper quantification [5]. The remaining ambiguity, which reflects several isomeric lipid species, needs to be made transparent. Data repositories and databases for mass spectrometry metabolomics data mostly also support lipidomics data; however, the most up to date lipid nomenclature should be used when submitting study data. The human metabolome database (HMDB) [127] cross-links chemical data on small molecules in the human body, including lipid data, to mass spectral evidence, clinical and biological data. MassBank [128] provides a reference spectrum database for the life sciences, covering many small molecule chemicals of different origin as well as small molecule standards, acquired on a wide variety of different MS instrumentation. Submissions are provided in the MassBank record format. MassBank, such as HMDB provides cross-links to other resources, such as KEGG [129], PubChem [130], ChEBI and LIPID MAPS.

The Global Natural Products Social Network GNPS [131] provides a novel way to interrelate MS^2^ signals based on graph-based proximity and allows propagation of identifications to previously unlabeled features. It supports import from Metabolomics Workbench projects and via the mzTab-M format for ad-hoc analyses. For ion mobility, the CCS Compendium [132] provides an online database with CCS values for chemical standards measured on drift tube ion mobility mass spectrometry devices. It maps its entries to the ClassyFire Chemical Ontology. The Panomics CCS database [133] for metabolites and xenobiotics integrates CCS values of metabolites and lipids with pathway information.

The MetaboLights [13] repository supports submission of metabolomics study metadata and raw MS and NMR data. It uses a specialized submission format based on the ‘Investigation-Study-Assay’ tab-separated (ISA-tab) text format linking multiple files, with support for MS, chromatography as well as NMR data, preferably in mzML and imzML formats, but other formats are also allowed.

Metabolomics Workbench [14,35] is another repository for metabolomics study metadata and raw MS and NMR data. It uses a single, text-based, tab-separated format (mwTab) and requires MS and NMR data to be provided preferably in mzML, mzXML or CDF formats. Both ISA-tab and mwTab contain information about the study design, study factors, samples, analytical procedures, parameters and software used for MS or NMR acquisition and subsequent data processing and support both quantitative as well as qualitative reporting of lipids and small molecules in general.

Metabolonote [134] is a wiki-based repository for metabolomics study metadata. MS data is referenced from MassBank or MassBase and other external repositories. Plant related datasets in Metabolonote are cross-linked to the Plant Genome Database Japan.

A meta repository that indexes studies from multiple repositories is MetabolomeXchange [135]. It allows for the browsing of study metadata provided from each repository and links out to the original datasets.

For imaging mass spectrometry, METASPACE [136] expects submissions in the imzML and ibd file formats and requests separate input of sample and processing information during the submission process. One drawback in repositories at the present point in time is that they often try to link identifications to InChI identifiers or other molecular representations with fully resolved structures, e.g., SMILES, which may not be warranted by the available mass spectrometric evidence. Thus, repositories should also be motivated to link to resources that support intermediate levels of structural resolution, such as LIPID MAPS and SwissLipids.

An important component for the integration of quantitative and qualitative data on lipids and lipid mediators is proper representation in pathway models. LimeMap [137] provides such a mapping for lipid mediators, associating them to interaction partners, such as enzymes, ion channels and receptors, based on mouse model gene names and orthologues in humans and rats.

Finally, an important and incredibly comprehensive resource for general background information on lipid biochemistry in mass spectrometry of fatty acid derivatives, complemented by short reviews on recently published work in the lipidomics field, is provided by the LipidWeb [138] blog.

## 7. Discussion

Open data standards for the recording and sharing of raw, intermediate and experimental results and their respective metadata play a crucial role in today’s interconnected, multidisciplinary omics sciences. The FAIR principles for research data handling and stewardship in the life sciences have summarized the availability and re-usability of scientific data as one of the crucial points for a higher return on investment of research results that would otherwise be inaccessible and, in the course of time, lost to digital amnesia (file corruption, interoperability issues, hardware failures). Having standardized formats simplifies submissions into FAIR data repositories and therefore, helps to prevent such issues. One further important application of standardized data formats is the evaluation of newly developed methods and algorithms on established “gold standard” data, as well as the general integration of different tools into larger bioinformatics pipelines. Using graphical workflow management systems, such as Galaxy [139] and KNIME [140], or programmatic/declarative workflow systems, such as Snakemake [141], Nextflow [142] or CWL [143], enable the creation and sharing of reproducible data workflows that have all parameters to individual processing steps defined and documented. If, in addition to the derived identification and quantification data, raw data is also made available, new methods can also be applied to reanalyze historical data to yield new results.

However, this is only possible if sufficient metadata about the measured samples, the underlying study design and the MS technology is also made available. For such reporting of workflow systems, it is crucial to ensure repeatability and reproducibility; thus, tools and databases should be continually maintained and need to have proper versioned releases, defined licensing terms and preferably, easy access to the source code to enable fair, credited reuse and adaptation. This currently seems to be the case for most of the tools and databases we reviewed, however, around a quarter to a third of them do not meet at least one of those requirements.

The Proteomics Standards Initiative (PSI) and other special interest groups in the omics sciences, e.g., the Metabolomics Standards Initiative (MSI) and the Lipidomics Standards Initiative (LSI), try to address the issues of reporting scientific findings in a way that is reproducible, findable and interpretable. They addressed these issues by defining recommendations for minimum information required for reporting experimental results. Moreover, they defined extensible data formats that are flexible to be extended for special use-cases, but still rigid enough to report essential information. To let the relatively young lipidomics community profit from the long-standing pioneering work of the proteomics and metabolomics communities, we investigated whether the already available HUPO PSI standard data formats can be used for lipidomics data and whether existing free lipidomics software tools already support them.

We did not find any technical obstacles for the direct usage of mzML for lipidomics raw data and peak lists. Hence, the complete adoption of mzML by the lipidomics community is technically unproblematic and it would be advantageous for the lipidomics community to profit from existing software tools for this format. Consequently, one could expect that most lipidomics software tools already use mzML as an import format.

However, a remarkable finding of this paper is the still prevalent use of mzXML. Since mzML was introduced several years ago as a unifying successor and replacement of mzData and mzXML and is supported by vendors and open-source software, the low acceptance in the lipidomics software field demands further effort in advocating it as a standard format. Thus, we clearly recommend all lipidomics software developers to support mzML for MS data import to simplify and unify processing, integration and interaction between different tools and workflows. In contrast to mzML, several different technical obstacles prevent the direct usage of mzIdentML for lipidomics identification results. Especially, one cannot report lipidomics results analogously to proteomics results, since lipidomics workflows are usually based on rules that link fragments to specific head-groups and fatty acid chain fragments.

Those are, in turn, backed up by analytical and mass spectrometric evidence from different levels of fragmentation. Based on the combined evidence they can determine the lipid species or more intricate structural features. Those issues, however, are addressed by the mzTab-M format, which can encode features, identification evidence and final quantities together with the necessary metadata in one file.

In contrast to proteomics [144], there is currently no widely accepted false discovery rate (FDR) concept available for lipidomics [140], although there are some attempts for significance estimation [141] in metabolomics and also, some of the tools presented here devise their own approaches.

For lipidomics, it is essential to report enough information to uniquely describe the knowledge about identified lipids [145], e.g., the structural level at which they were identified. This defines the need for a nomenclature for a unique definition and ontological classification of lipids, their structures and fragment ions. An early proposal was the three-level classification scheme introduced by the LIPID MAPS consortium. However, it lacked concepts for describing further intermediate levels of identification that are accessible with current MS technology with progressively more structural information. These levels have been incorporated by the more detailed nomenclature, introduced in [123] and its recent update [126], that are already supported by LIPID MAPS, RefMet and Goslin.

Finally, we assessed that mzTab can already be used as the output or end file format for lipidomics data. However, in its first version (mzTab 1.0), it could only report summary data without richer information to back up the identification and quantification results with evidence. The mzTab-M 2.0 format for metabolomics addresses these issues and provides a basis for lipid-specific extensions through additional columns, metadata and semantic validation rules for specific lipidomics workflows. These extensions and customizations would warrant a backwards-compatible mzTab-L, based on mzTab-M, that would be usable as a standardized data reporting and exchange format and would also be a proper format for the deposition of lipidomics results in public repositories. Support for mzTab-M has already been implemented in LDA2, MS-DIAL, GNPS and MetaboAnalyst and will be supported by the upcoming releases of MZmine 3.

The lipidomics community can further benefit from the new standardization developments within the other mass spectrometry-based communities, e.g., MAGE-TAB-Proteomics [146]. A HUPO-PSI format that has recently been developed can be a template for an analogous lipidomics-specific file format. MAGE-TAB-Proteomics describes the metadata of the samples of a dataset and their association with the dataset files, allowing their full understanding or reanalysis. Consequently, the interpretability and reusability of lipidomics data would greatly benefit from alignment with MAGE-TAB in a lipidomics-specific format.

## 8. Conclusions

The current state of bioinformatics tools, data formats and resources in lipidomics is rapidly evolving. Thus, we recommend that in the short term, the lipidomics community, together with established bodies such as the PSI and LSI, should join forces to further standardize the naming and reporting of lipidomics data. We suggest that the LSI and all interested parties should continue the discussions and efforts regarding lipidomics-specific extensions and updates of the PSI formats to have an exhaustive set of proven standard data formats, enabling the compliance with the FAIR data principles and allowing easier data integration across mass spectrometry experiments, for example with the proteomics and metabolomics fields and across domains, such as the human health and natural product communities.

To simplify tool accessibility, maintenance and reusability, developers should publish their source code under an open-source license in publicly available source code repositories such as GitHub or GitLab that allow for easy collaboration and feedback, i.e., to contact the developers in case of missing features or bugs. Building a community around these tools and resources will also help to counter the problems associated with continued maintenance and updates that many tools suffer from after the initial developer has moved on or after the project funding has ceased.

In the long term and with reasonable adoption by lipidomics tool developers, these efforts could lead to the much simpler exchange and reuse of both lipidomics data and tools, as well as an overall improved data quality in the field that will be strengthened by providing citable and accessible results along with raw data for secondary reuse and scientific benefit. Further, these standardization efforts will, in the long term, enable high-throughput application of lipidomics and simplify integration with data from other omics domains to pave the way for applications in systems biology and precision medicine.

## Figures and Tables

**Figure 1 metabolites-12-00584-f001:**
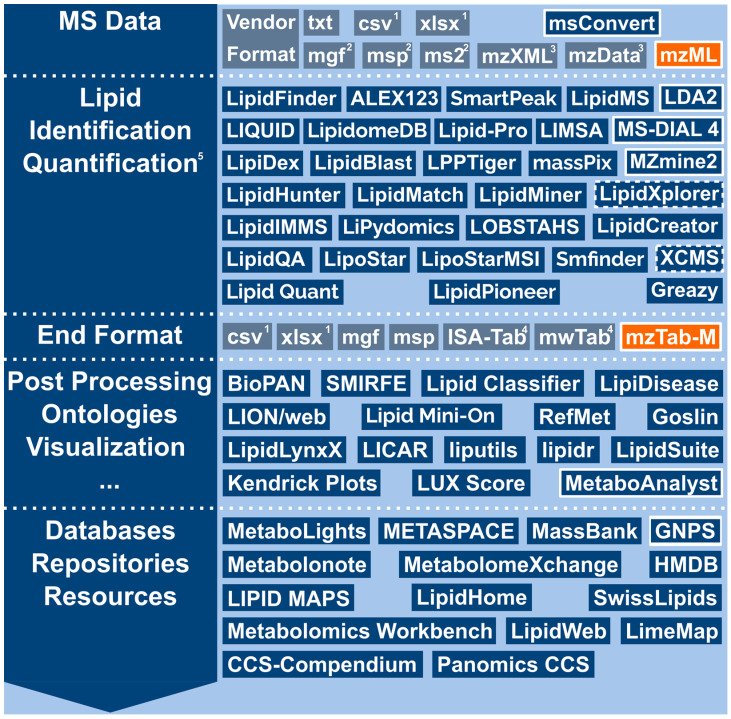
Data flow for different lipidomics software tools. From top to bottom, the supported ‘data format paths’ of selected lipidomics software tools that support at least one PSI standard data format are highlighted with a solid white outline. Those with planned or upcoming support are highlighted with a dashed white outline. The various software tools are represented by blue rectangles with the tool name in white. Data formats are represented by gray rectangles. On the right, the theoretically possible data flow between the PSI standard data formats, mzML and mzTab-M, is depicted (orange background) via supporting software and repositories/databases (white rectangle outline). In this figure, ‘Vendor Format’ stands for various proprietary, vendor-specific raw data formats (i.e., Thermo Fisher .raw, Agilent .d, ABSciex .wiff and Waters .raw), which can be converted to mzML (among other formats) using msConvert. (1) ‘csv/xlsx’ represents non-standardized output formats such as Microsoft XLSX, comma/tab separated text file formats and HTML. (2) ‘mgf/msp/ms2‘ are text-based formats that encode mass spectral data but generally do not have a strongly defined metadata schema. (3) ‘mzXML/mzData’ represent the legacy raw data and peak list formats mzXML and mzData. (4) ISA-Tab (used by MetaboLights) and mwTab (used by Metabolomics Workbench) are text-based, tabular data formats based on a defined metadata model, which simplifies validation and tooling. (5) Only some tools support quantification. See Table 1 for details.

**Table 1 metabolites-12-00584-t001:** Overview of software for lipid identification from mass spectrometry. Abbreviations: U: Untargeted, T: Targeted, C: Chromatography, CE: Capillary Electrophoresis, IM: Ion Mobility, DI: Direct Infusion (Shotgun), I: Imaging. $: targeted includes Selected Reaction and Multiple Reaction Monitoring (MRM), untargeted includes DDA and DIA approaches. *: Only the most important ones relevant to this review. All tools use some form of configuration file format, e.g., text-based (TXT) or other formats for libraries or fragmentation rules. Workflow assignment designates the primary workflow a tool was designed for and this was stated by the authors; others may be available. We use direct infusion as a more generic synonym for what is usually referred to as ‘shotgun lipidomics’. Comma-separated values (CSV) is a tabular, spreadsheet-like format. If tab characters are used as separators, the format is TSV. Hypertext markup language (HTML) is a format viewable with an internet browser. XLSX: MS office XML-based spreadsheet format. MSP: NIST mass spectral library format. MGF: Mascot Generic Format. BLIB: Binary mass spectral library format. PDF: Portable Document Format. #: rule-based validation often includes spectral scores, ratios and thresholds, scores denote spectral similarity functions, such as the commonly used dot product/cosine variants. Remarks: (1) The software is no longer available. (2) Lipid class separation chromatography, e.g., HILIC or supercritical fluid chromatography. (3) XCMS input recommended, LIPID MAPS class assignment of suspect ions. (3) Software is provided as a web application without further information. (4) Supports phospholipids only. (5) XCMS input recommended, LIPID MAPS class assignment of suspect ions. (6) After release 3.0, LipidMatch is available as LipidMatch Flow (latest version 3.5, but without source code). (7) Supports oxidized phospholipids only. (8) Identification and quantification use other tools’ methods. (9) The source code is provided for download, but no code license is defined.

Workflow $	Name	Handling	MS *	Identification #	Quant	Input	Output	Last Release	Open-Source	License	Programming Language
T	LIMSA	C, DI	MS^1^, MS^2^	Compound/Fragment library	yes	XLSX, CSV, HTML	NA	2006	NA (1)	GPL v3	C++, VBA, Excel
T	LipidomeDB	DI, C	MS^1^, MS^2^	*m*/*z* Library + Transitions + rule-based	yes	XLSX	XLSX, HTML	2019	no	NA	Java
T	LipidQuant	C (2)	MS^1^	*m*/*z* library + rule-based	yes	TXT	XLSX	2021	yes	CC-BY 4	VBA, Excel
U	ALEX and ALEX 123	DI	MS^1^, MS^2^, MS^3^	Manual	no	manual input of parameters	HTML	2017	no	NA	NA (3)
U	Greazy (4)	C, DI	MS^1^, MS^2^	Fragment/Spectral Library + score	no	vendor, mzML	mzTab (via LipidLama)	2022	yes	Apache v2	C#
U	LDA2	C	MS^1^, MS^2^	Rule-based	yes	mzML, TXT	XLSX, mzTab-M	2021	yes	GPL v3	Java
U	LipidBlast	C	MS^1^, MS^2^	Spectral Library + score	no	MSP, MGF, XLSX	MGF, XLSX	2014	yes	CC-BY	EXCEL
U	LipiDex	C	MS^1^, MS^2^	Spectral Library + rule-based	yes	MGF, mzXML, CSV	CSV	2018	yes	MIT	Java
U	LipidFinder	C	MS^1^	Rule-based, LMSD	no	CSV, JSON (5)	PDF	2021	yes	MIT	Python
U	LipidHunter (4)	C, DI	MS^1^, MS^2^	Rule-based	yes	mzML, XLSX, TXT	XLSX, HTML, TXT	2020	yes	GPL v2, Proprietary	Python
U	LipidIMMS	C, IM	MS^1^ + CCS, MS^2^	CCS Library + Spectral Library + score	no	MSP, MGF	CSV, HTML	2020	no	NA	NA (3)
U	LipidMatch (6)	C, I, DI	MS^1^, MS^2^, MSE/DIA	Compound/Fragment library + rule-based	yes	CSV, MS2 (ProteoWizard)	CSV	2020	yes	CC BY 4.0	R
U	LipidMiner	C	MS^1^, MS^2^	Compound/Fragment library + rule-based	yes	raw	XLSX, CSV	2014	no	NA	C#, Python
U	LipidMS	C	MS^1^, MS^2^, MSE/DIA	Compound/Fragment library + rule-based	yes	mzXML, CSV	CSV	2022	yes	GPL v3	R
U	Lipid-Pro	C	MSE/DIA	Compound/Fragment library	yes	CSV	XLSX, TXT	2015	no	Proprietary	C#
U	LipidXplorer	DI	MS^1^, MS^2^, MS^3^	Rule based	no	mzML (MS^1^ + MS^2^)	CSV, HTML	2019	yes	GPL v2	Python
U	LiPydomics	C, IM	MS^1^	CCS Library + *m*/*z* Library + HILIC RT Library + rule-based	yes	CSV	XLSX	2021	yes	MIT	Python
U	LIQUID	C	MS^1^, MS^2^	Spectral Library + rule-based	yes	RAW, mzML	TSV, mzTab, MSP	2021	yes	Apache v2	C#
U	LOBSTAHS	C	MS^1^	Spectral Library + rule-based	yes	mzML, mzXML, mzData, CSV	XLSX, CSV	2021	yes	GPL v3	R
U	LPPTiger (7)	C	MS^1^, MS^2^	Spectral Library + score	yes	mzML, XLSX, TXT	XLSX, HTML	2021	yes	GPL v2, Proprietary	Python
U	MassPix	I	MS^1^	*m*/*z* Library + rule-based	no	imzML	CSV	2017	yes	NA	R
U	MS-DIAL 4	C, CE, IM	MS^1^, MS^2^, MSE/DIA	Spectral Library + rule-based	yes	vendor, mzML	CSV, mzTab-M, XLSX	2022	yes	GPL v3	C#
U	MZmine 2	C	MS^1^, MS^2^	Spectral Library + rule-based	yes	vendor, mzML, mzXML, mzData, CSV, mzTab, XML	CSV, mzTab, XML	2019	yes	GPL v2	Java
U	XCMS	C	MS^1^, MS^2^	Spectral Library + score	yes	mzML, mzXML, netCDF	CSV	2021	yes	GPL v2	R, C
T + U	LipidCreator and Skyline	C	MS^1^, MS^2^, MSE/DIA	Fragment/Spectral Library + score (8)	yes (8)	vendor, mzML (MS^1^ + MS^2^)	XLSX, CSV, BLIB	2021	yes	MIT	C#
T + U	LipidPioneer	C	MS^1^, MS^2^	Compound/*m*/*z* Library (8)	yes (8)	XLSX	XLSX	2017	yes (9)	NA	VBA, Excel
T + U	LipidQA	DI	MS^1^, MS^2^	Spectral Library + score	yes	vendor (Thermo, Waters)	CSV	2007	NA (1)	NA	Visual C++
T + U	LipoStar	C, IM	MS^1^, MS^2^, MSE/DIA	Compound/Fragment library + rule-based validation	yes	vendor	CSV	2022	no	Proprietary	C#
T + U	LipoStarMSI	DI, I	MS^1^, MS^2^	Spectral Library + rule based	yes	vendor (Bruker, Waters), imzML	CSV	2020	no	Proprietary	C#
T + U	SmartPeak	C	MS^1^, MS^2^	Transitions + rule-based	yes	mzML, CSV	mzTab, XML, CSV	2022	yes	MIT	C++, Python
T + U	Smfinder	C	MS^1^, MS^2^	Spectral Library + score	yes	mzML, mzXML	XLSX, TXT	2020	yes (9)	NA	Python, R, C++

**Table 2 metabolites-12-00584-t002:** Libraries and web applications for Pathway analysis, ontology mapping/classification, enrichment analysis, post-processing, visualization and statistical analysis. Remarks: (1) Library Rodin is used by the web application. (2) From molecular formulas. (3) Figshare id. (4) Based on lipid structural features. (5) Part of Bioconductor release 3.14. (6) Only the R package is open-source. (7) R package MetaboAnalystR 3.2 (2021). (8) Part of MZmine 2. (9) MZmine 3 release is planned for 2022.

Category	Name	Type	Open Source	License	Programming Language	Last Release	Version
Ontology, Enrichment	Lipid Mini-On	Web application, Library (1)	yes	BSD 2-Clause	R	2019	0.1.43
Ontology, Enrichment	LION/web	Web application	yes	GPL v3	R	2020	NA
Ontology, Enrichment	LipiDisease	Web application	no	NA	R	2021	NA
Ontology, Classification (2)	SMIRFE	Library	yes	NA	Python	2020	187eb261983b6d0aca1c (3)
Ontology, Classification (4)	Lipid Classifier	Library	yes	A-GPL v3	Ruby	2014	0.0.0.1
Ontology, Enrichment, Pathway Analysis	BioPAN	Web application	no	GPL v3	PHP, R, HTML, JavaScript	2020	NA
Post-Processing	Goslin	Web application, Library	yes	MIT, Apache v2	C++, C#, Java, Python, R	2022	2.0
Post-Processing	LipidLynxX	Web application, Library	yes	GPL v3	Python	2020	0.9.24
Post-Processing	RefMet	Web application	no	NA	PHP, R	2021	NA
Post-Processing	LICAR	Web application	yes	MIT	R	2021	1.0
Statistical Analysis, Visualization	lipidr	Library	yes	MIT	R	2021	2.8.1 (5)
Statistical Analysis, Visualization	LipidSuite	Web application	no	NA	R	2021	1
Statistical Analysis, Visualization	liputils	Library	yes	GPL v3	Python	2021	0.16.2
Statistical Analysis, Visualization	MetaboAnalyst	Web application, Library	no (6)	GPL v2	Java, R (7)	2021	5.0
Visualization	Kendrick mass-defect plots	Library (8)	yes	GPL v2	Java	2019 (9)	2.53
Statistical Analysis, Visualization	LUX Score	Web application, application	yes	Apache v2	Perl, R, Python	2018	1.0.1

**Table 3 metabolites-12-00584-t003:** Overview of databases and resources for lipidomics grouped by classification, specific support for lipids, general availability of lipid structures, support for different levels of structural resolution (shorthand notation), main type of lipid ontology supported, availability of mass spectral data, availability and cross-linking of biochemical reaction data and curation model. Remarks: (1) kingdom, superclass, class, subclass. (2) internal and through MassIVE. (3) via integration with multiple tools. (4) via MassIVE and other public repositories. (5) via search. (6) local and linked via SPLASH [119] to MONA, MassBank. (7) via search and shorthand abbreviation. (8) Original and Liebisch 2020. (9) via Metabolomics Workbench. (10) GP and GL only. (11) based on the submission format (Mass Bank format). (12) via reference to spectral data. (13) based on submission format ISA-Table. (14) based on submission format mwTab. (15) others are available, e.g., LIPID MAPS, SwissLipids. (16) metabolic pathways of lipid mediators. (17) not necessarily machine readable.

Category	Name	Main Purpose	Lipid Specific	Lipid Structures	Structural Levels	Ontology	Spectral Data	Biochemical Reaction Data	Curation
Database	CCS-Compendium	Compendium of experimentally acquired Collisional Cross Section (Ion Mobility) data from molecular standards acquired on drift tube instruments	yes	yes	yes (1)	ClassyFire/ChemOnt	no	no	manual
Database	Panomics CCS	Collisional Cross Section (Ion Mobility) Database for Metabolites and Xenobiotics acquired on drift tube instruments	no	yes	no	no	no	yes	manual
Database	GNPS	Knowledge base for raw, processed or annotated fragmentation mass spectrometry data	no	yes	no	-	yes (2)	yes (3)	no (4)
Database	HMDB	Curated database of small molecule metabolites found in the human body	no	yes	yes (5)	ClassyFire/ChemOnt	yes (6)	yes	manual
Database	LIPID MAPS	Curated portal for LIPID MAPS lipid classification, experimentally determined structures, in-silico combinatorial structures and other lipid resources	yes	yes	yes (7)	LIPID MAPS (8)	yes (9)	yes	manual
Database	LipidHome	In-silico generated theoretical lipid structures	yes	yes (10)	no	Liebisch 2013	no	no	manual
Database	SwissLipids	Curated database of lipid structures with experimental evidence and integration with biological knowledge and models	yes	yes	yes	Liebisch 2013	no	yes	manual
Repository	MassBank	Curated database of mass spectrometry reference spectra	no	no	no	-	yes	no	manual (11)
Repository	MetaboLights	Repository for metabolomics data (MS and Nuclear Magnetic Resonance (NMR)) and metadata	no	yes	no	ChEBI	yes (12)	no	manual (13)
Repository	Metabolomics Workbench	Repository for metabolomics data (MS and NMR) and metadata	no	yes	yes	RefMet	yes	no	manual (14)
Repository	Metabolonote	Wiki-based repository for metabolomics metadata	no	no	no	-	yes (12)	no	manual
Repository	MetabolomeXchange	Aggregator of metabolomics metadata from MetaboLights, Metabolomics Workbench, Metabolonote and Metabolomic Repository Bordeaux	no	no	no	-	no	no	no
Repository	METASPACE	Repository for imaging mass spectrometry for metabolomics	no	yes	no	HMDB/ClassyFire/ChemOnt (15)	yes	no	manual
Resource	LimeMap	Curated CellDesigner XML and Vanted GML graph of lipid mediator pathways	yes	no	no (15)	-	no	yes (16)	manual
Resource	LipidWeb	Literature review and biochemistry of lipids	yes	yes (17)	no	-	yes (17)	yes (17)	manual

## Data Availability

A curated collection of the bioinformatics tools, databases and resources is available at GitHub under the terms of the Creative Commons Attribution-Share Alike 4.0 International License—CC BY-SA 4.0 following the popular “Awesome collection” approach: https://github.com/lifs-tools/awesome-lipidomics (accessed on 24 May 2022).

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
