# Peer review of "A Current Encyclopedia of Bioinformatics Tools, Data Formats and Resources for Mass Spectrometry Lipidomics"

_metabolites, 2022, doi:10.3390/metabo12070584_

Round 1

Reviewer 1 Report

The review article entitled " Bioinformatics Tools, Data Formats and Resources for Mass Spectrometry Lipidomics" is a well written review with detailed information on various available tools and resources for the identification and processing of lipidomics data generated by MS. The manuscript covered up to date literature and also discussed the potential future prospectives. 

However, I have few concerns

1. Authors should to include MetabolomeXchange in the resources section of the manuscript. It is an international data aggregration and notification tools for metabolomics data. Also, Metabolonote is a database used to manage metadata and authors should discuss about its importance in the manuscript.

2. Authors should define the full forms of the abbreviation when they are mentioned in the article at first appearance. For example, line 36, MS; Line 37, HUPO-PSI etc.,

Reviewer 2 Report

I suggest you to add a separate section of "materials and methods", in which describe the search methodology and the eligibility criteria, now reported in the Introduction from line 113 onwards.

There is a duplicate in the section number referring to line 317 and 450.

I suggest you to increase the Conclusions section.

Reviewer 3 Report

“Bioinformatics Tools, Data Formats and Resources for Mass Spectrometry Lipidomics” provides a comprehensive and thorough investigation of software tools for lipidomics. It is surprisingly thorough, covering 57 software / databases / apps and the open-source file formats used. The authors are commended for this work. For this reason, I may rename the title to “A Current Encylopedia of Open-Source Lipidomics Software, Databases, and File Formats” or something Along the lines to attract readers to use this as a comprehensive one-stop shop to gain some brief insights into all that is available. It also reads like a list… so encyclopedia might be appropriate terminology.

The tables and lists are very informative, but I find the introduction, and paragraphs between lists and descriptions to have no flow. For example, in the introduction topics jump around without clear flow, it starts with ion mobility, moves to QqQ and targeted, then to High-Res targeted, then to DDA and DIA, then to tandem mass spectrometry…

I might focus more in the introduction on why software is needed in lipidomics, what steps it covers and how that reflects data-acquisition methods, and what are the challenges that software in lipidomics as solved and what is needed in the future (briefly). A figure going through data-processing options (big picture), visualizations, network analysis, database storage and other steps and needs for software in an organized fashion could be of interest. Also maybe defining FAIR and some of the lenses for which you are looking at the software through: adaptability, open-source community-based development and what is needed (e.g., the discussion about file format), etc. what is needed for this to all happen successfully?

I find of most interest the unique and new discussion on file format, and really enjoy the discussion on a file format which includes all the MS LC etc. information AS WELL AS annotations, features, etc. So this could be highly in the abstract if you would like.

Finally, while I applaud the request for using the same file type, especially with results integrated as it would be wonderful to be able to integrate results, spectra, etc. from one open-source software to another. The challenge is getting open-source developers who are exhausted of resources and time to change things that are already working.

Specific Edits:

“Decreasing acquisition time”

I am not sure this is true, to get good separation in LC-HRMS/MS longer gradient are needed, methods range from 10 minutes to 3 hours, and in reverse phase the longer the gradient the better separation. When has this improved? Actually, with intelligent acquisition methods and new methods, MORE not less injections are required.

LipidMatch (including LipidMatch Flow) is written in R and C#, and because it integrates MZMine and MSConvert, it also deals with java, but we can leave that out.  LipidMatch 4.0 Is in development and by the time this is published will be released in 2022. So, keep eyes out before publication to update table.

Greazy: I think an important point here is that this software is not rule-based and the probability scores are based on the number of fragments observed for a specific lipid. Also, the libraries are generated in-silico based on user input (e.g., number of carbons unsaturation, even and odd versus just even, etc.) which also makes it unique. Also, there libraries have a very high number of fragments. This software is no longer supported, as the developer retired.

LipidBlast libraries were integrated into MS-DIAL and MS-DIAL can be seen as the evolution and further development of LipidBlast. Therefore, you might include LipidBlast and part of MS-DIAL description as anyone using LipidBlast would be better to use MS-DIAL which as integrated this and is still developed. They were developed in the same lab (Fein lab) and then Hiroshi took it on at Riken and has continued. This is updated continuously and is probably the most widely used and cited software in lipidomic.

LipidMatch: Feature tables are accepted as csv, MS/MS files are accepted as .ms2. In the LipidMatch Flow version vendor files are accepted without the need for users to convert files (employs MSConvert in the background). Similar to MZMine now. Check for new release soon, LipidMatch 4.0 will also publish files (csv) which can be imported into Microsoft PowerBI (free) for visualization. Visualization is another aspect that may want to be covered. MS-DIAL is one of the only other free software with a good visualization platform. LipidMatch incorporates MS-DIAL libraries as well (via collaboration, yay open-source!) so pseudo has the same number.

LIQUID: has a unique scoring based on quartiles of intensity to include intensity for scoring without having to have extremely accurate in-silico predictions.

LOBSTAHS: Mainly uses adduct order of intensity (most to least) to identify lipids which is very unique approach.

MS-DIAL: Also has retention time prediction algorithms which is unique, visualizations, the largest lipidomics library…

MZMine lipidomics apply or atleast include LipidMatch libraries I believe although may be wrong.

LipidPioneer: Generally it generates formulas and a range of adduct masses observed in MS, these can be used for inclusion or exclusion lists as mentioned, but also have a plurality of uses, including for generating EICs while running the instrument of known or expected compounds (e.g., generation of layouts) for QA/QC and as a first check to see if a compound of interest is in a sample (of course MS/MS is further required for validation). 
